# The Adaptability of Chromosomal Instability in Cancer Therapy and Resistance

**DOI:** 10.3390/ijms24010245

**Published:** 2022-12-23

**Authors:** Vinicio Carloni, Elisa Morganti, Andrea Galli, Antonio Mazzocca

**Affiliations:** 1Department of Experimental and Clinical Biomedical Sciences, University of Florence, 50121 Florence, Italy; 2Interdisciplinary Department of Medicine, University of Bari-School of Medicine, 70121 Bari, Italy

**Keywords:** DNA damage response, DNA damage, epigenetic changes, immunotherapy, immune evasion

## Abstract

Variation in chromosome structure is a central source of DNA damage and DNA damage response, together representinga major hallmark of chromosomal instability. Cancer cells under selective pressure of therapy use DNA damage and DNA damage response to produce newfunctional assets as an evolutionary mechanism. Recent efforts to understand DNA damage/chromosomal instability and elucidate its role in initiation or progression of cancer have also disclosed its vulnerabilities represented by inappropriate DNA damage response, chromatin changes, andinflammation. Understanding these vulnerabilities can provide important clues for predicting treatment response and for the development of novel strategies that prevent the emergence of therapy resistant tumors.

## 1. Introduction

Treatment of cancer usually includes surgery, radiotherapy, or chemotherapy. The activity of radiotherapy and most chemotherapy drugs induces DNA damage in proliferating tumor cells. For example, platinum drugs that are effective against cancer cells form DNA adducts. Other drugs, such as anthracyclines (e.g., doxorubicin) induce DNA damage by inhibiting DNA topoisomerases and promoting the production of mitochondrial reactive oxygen species (ROS). In normal cells, DNA damage can be immediately recognized by a DNA damage response (DDR), which activates checkpoints for cell-cycle arrest and DNA repair [1,2]. In cancer cells, DNA damage per se may be responsible for defective chromosome segregation and chromosomal instability (CIN) or the anomalous binding of DDR proteins to DNA or chromatin proteins. Many DDR proteins have been uncovered, and, depending on the type of damage, several DNA repair pathways have been characterizedand extensively reviewed previously [3,4,5]. The challenge is now to figure out how the different DNA repair pathways interact with each other to modulate DNA damage/CIN. The relationships among DNA damage, DDR signaling, and CIN adaptability mechanisms affecting the effectiveness of chemotherapy and/or immunotherapyare addressed in this review.

## 2. DNA Damage and Chromosomal Instability

Exogenous and endogenous insults can cause DNA damage. Ionizing radiation (IR) and chemotherapeutic agents, particularly lethal to cancer cells, cause DNA double-strand breaks (DSBs); cell metabolism products (e.g., reactive oxygen species) can cause DSBs as well, but it is also clear that stress replication and chromosome mis-segregation provoke DNA damage.Moreover, chromosomes containing damaged DNA segregate inappropriately because sister chromatids are still connected by DNA and/or DNA-proteins during the division of DNA into daughter cells [6,7,8]. In anaphase, defective chromosome segregation can produce lagging chromosomes or bridge chromosomes.Indeed, errors in the function of the mitotic spindle can produce lagging chromosomes such that the entire chromosome cannot be properly separated due to the connection with the microtubules derived from the opposite spindle poles [9,10,11]. Anaphase spindles working inappropriately may also induce bridged chromatin in such a way that DNA is extended to opposite spindle poles from the same chromosome or non-disjoined sister chromatids.Chromatin bridges are hallmarks of structural chromosomal instability.Noteworthily, lagging chromosomes and chromatin bridges often do not catch up with the other chromosomes to be introduced in the reforming nucleus. This chromatin self-organizes into micronuclei surrounded by a nuclear membrane (Figure 1). The level of DNA damage in micronuclei is high, and the repair of this damage leads to extensive gene rearrangements [12].

## 3. DDR Components Are Targets of Chemotherapy Drugs

DNA is subjected to different forms of damage, and the type of damage incurred dictates the subsequent repair mechanism. When damage is limited to one strand, the lesion is often repaired by excision of the damaged base followed by DNA synthesis using the opposite strandas a template.These single-stranded forms of DNA damage following chemical base modifications or misincorporation of ribonucleotides are frequent and are efficiently corrected by the cell using either base excision repair or nucleotide excision repair mechanisms. However, photodimers and intra- and interstrand crosslinks, among other obstacles, can block replication and may result in DNAdouble-strand breaks (DSBs) [5]. Moreover, single-strand nicks canbe converted into one-ended DSBs upon replication fork passage. DSBs can arise directly from many additional sources, including IR, the excision of transposable elements, failures of type II topoisomerases, and the action of site-specific endonucleases. Vertebrate cells suffer a dozen or more chromatid breaks every replication cycle. This is evident from the high frequency of chromatid breaks when the key Rad51 repair protein is depleted from chicken DT40 cells and the frequency of sister-chromatid exchanges in humans when the BLM helicase is absent [6,7]. Unlike single-stranded DNA damage, a DSB cannot be repaired by using the complementary strand as a template; instead, one of two repair mechanisms isemployed: nonhomologous end joining (NHEJ) or homologous recombination (HR).These pathways are complementary and operate optimally under different circumstances. HR requires the presence of a homologous template, usually a sister chromatid, which allows accurate repair of DSBs in the S and G2 phases of the cell cycle. In contrast, NHEJ can operate throughout the cell cycle without the need for template DNA and is often mutagenic because deletions or insertions can be induced at sites of repair. Site-specific endonucleases, which causeDSBs at specific locations within the genome, have been used to detail the mechanisms behind HR and NHEJ.

Failure to repair DSBs causes genomic instability and can lead to tumorigenesis and other age-related diseases. Upon DSB induction, cells activate a DDR that comprises two major stages: initial sensing of DNA breaks followed by downstream events leading to cell-cycle arrest, DNA damage repair, and subsequent cell-cycle resumption. In the interphase cell cycle, the DDR pathways have been well characterized, but their function has remained unknown in mitosis for a long time. Recently, it has become clear that many proteins involved in DNA repair are also required for the progression of normal mitosis and faithful chromosome segregation. DDR signaling is initiated by Mre11–Rad50–Nbs1 (MRN) and Ku70–Ku80 complexes with recruitment ofATR/CHK1 and ATM/CHK2, which are activated by DNA damage caused by replication stress and chromosome mis-segregation, respectively. Some components of DDR signaling can be co-opted during mitosis by cancer cells to further propagate chromosomal instability [13,14]. The central effector of DDR, CHK2, can self-activate when it is overexpressed; self-activation is associated with CHK2 threonine 68 and 387 phosphorylation [15], and CHK2 overexpression can exert counterproductive effects such as in liver cancer leading to spontaneous and trans-phosphorylation events that in turn result in increased mis-segregation of chromosomes, supporting the notion of a concentration-dependent condition in promoting its activation, e.g., CHK2 overexpression identifies a subclass of hepatocellular carcinoma with chromosomal instability [16,17]. There are important therapeutic implications for the role of defective-mitosis related DDR in propagating structural and numerical chromosomal aberrations in liver cancer cells [18,19]. First of all, it seems that the commonly used therapies that damage DNA, such as IR, cisplatin, doxorubicin, and epirubicin, exert cytotoxic effects at least in part by interfering with the segregation of chromosomes. Although cancer cells with abnormal chromosome function show high rates of chromosome mis-segregation, these rates need to be optimized.The genomic heterogeneity required for tumor evolution can support neither the low rate of chromosome mis-segregation, nor the excessively high rate that may be detrimental to cell viability [14,20,21]. This means that there is an evolutionary limitation on the rate of chromosome mis-segregation that maintains cell viability but that, at the same time, promotes the acquisition of chromosome aberrations [21]. This view is supported by a large number of clinical observations in which moderate rates of chromosomal instability were most likely to give rise to therapeutic resistance and poor prognosis; on the contrary, a favorable prognosis and higher likelihood of therapeutic response were associated with either extremely high or extremely low chromosome mis-segregation rates. Identifying mechanisms driving the adaptation to and tolerance of chromosome aberrations might reveal new strategies to damp the selection of drugs resistant cancer cells. These findings are crucial in the context of concerted efforts aimed at targeting DDR components. On the basis of these ideas, drugs that inhibit the components of the DNA damage response mechanism, such as single-strand break DNA repair enzyme poly(ADP-ribose) polymerase(PARP) inhibitor, have been developed. Indeed, BRCA1 or BRCA2 mutant cells are susceptible to PARP inhibitor therapy because of their defective HR pathway to repair DNA that normally compensates for the loss of PARP activity [22,23]. It is important to note that mutations in other HR genes also increase the vulnerability of cells to PARP inhibitors; this is known as the BRCAness concept [24]. For example, patients with prostate cancer with mutations in BRCA2, PALB2, FANCA, RAD51,or CHK2 genes are responsive to PARP inhibitors [25,26,27]. As such, PARP inhibitors may be efficacious fora much wider variety of tumors than those specifically lacking BRCA1 function. However, BRCA2 mutant tumors treated with PARP inhibitors have shown drug resistance caused by frame deletions of the BRCA2, which partially restore its DNA repair function, thereby allowing cancer cells to survive [28]. The intimate association between reduced HR and the sensitivity of such cells to DNA damaging agents and PARP inhibitors has been central to the original definition of BRCAness [24]. However, recent deep sequencing studies have revealed sporadic mutations in BRCA genes not only in breast and ovarian cancer, but also in liver, colon, and pancreatic adenocarcinoma [29]. Interestingly, certain types of cancer, such as Ewing’s sarcoma, do not harbor mutations in BRCA1 or BRCA2, but they still are sensitive to PARP inhibitors [30,31]. In addition, several other factors are known to modulate BRCA1/2 protein function and, thus, have altered responses to HR and PARP inhibitors. For example, hypoxic conditions reduce BRCA1 transcript expression via repressive E2F transcription factors and changes in histone methylation, resulting in significantly reduced HR without affecting NHEJ [32]. On the other hand, BRCA1-deficient mice die embryonically due to the accumulation of endogenous DNA damage which activates DDR; interestingly, abolishing DDR signaling increases tolerance and promote survival in these mice.Deletion of ATM or the gene encoding its effector CHK2 renders BRCA1-deleted mice viable. Along these lines, the finding that 53BP1 (p53 binding protein 1) inhibition in BRCA1-deficient cells leads to partial HR reactivation was corroborated by in vivo studies [33,34]. In a mouse BRCA1-mutant cancer model, tumors that are sensitive to PARP inhibitor olaparib acquire resistance following prolonged exposure to the drug. Silencing of 53BP1 expression is a major cause of drug resistance and tumor re-growth. This discovery suggests that the expression level of 53BP1 could be a predictor of the response of BRCA1-deficient tumors to PARP inhibitors [35,36]. Loss of 53BP1 in BRCA1-deficient cells restores, to some degree, homologous recombination in a manner that depends on the activation of end resection [37]. This interaction underscores an antagonism between BRCA1 and 53BP1, a conclusion supported by studies where BRCA1 and 53BP1 appear to compete for accumulation at DNA damage sites [38]. These findings suggest that controlling DNA end resections are necessary in DSB repair, with a direct impact on the therapeutic efficacy of PARP inhibitors. DNA end resection promotes repair via homologous recombination, whereas minimally processed ends are repaired by non homologous end joining. How 53BP1 shields DNA ends from nucleases has been an enduring mystery. The recent discovery of shieldin (SHLD), a four-subunit protein complex with single-stranded DNA-binding activity, brought to light a strong candidate for the ultimate effector of 53BP1-dependent end protection. Depleting any single subunit in various BRCA1-deficient cell lines suppresses their sensitivity to PARPi to a degree comparable to that of 53BP1 [39,40,41]. The potency of this effect was illustrated in vivo in experiments where BRCA1-null mammary tumor cells edited to mutate SHLD1 or SHLD2 were resistant to PARPi treatment [39]. In addition, expression levels of SHLD1 and SHLD2 correlated with PARPi sensitivity in patient-derived xenografts of BRCA1-null tumors. These findings suggest that shieldin mutations modify PARPi responses in BRCA1-mutated tumors.

## 4. Importance of DDR and Chromatin Remodeling in Therapeutics Response and Resistance

In response to therapeutic intervention, cancer cells can redirect DNA repair pathways. NHEJ, which is the prevalent DSB repair pathway in higher eukaryotes, essentially mediates the direct ligation of broken DNA ends and usually involves minimal DNA end processing. Changes in the balance between HR and NHEJ, for example, can alter the DNA repair-deficient tumor cell response to DSB-inducing drugs (Figure 2). Although there is no error in HR, NHEJ can cause mutations that are favorable for survival. Hence, many tumor resistances lacking HR can be explained by abnormal regulation of NHEJ activity. Tumors with HR defects, through the rewiring of DNA repair pathways, will thwart the therapeutic response. For instance, as aforementioned, loss of 53BP1 rescues cancer cells with BRCA1 deficiency, including HR impairment and chemotherapeutic sensitivity [35,36,42]. Similarly, overexpression of mutant 53BP1 inhibits NHEJ and stimulates HR in an XRCC4-dependent manner. A similar change from NHEJ to HR was also found in embryonic stem cells deficient in XRCC4 or other NHEJ proteins [43,44]. The blocking of NHEJ by Lig4 deletion did not, however, rescue BRCA1 mutant cells from genomic instability. The results indicate that the HR recovery caused by the loss of 53BP1 is an important factor in the chemotherapy response [36,42]. Therefore, since the abnormal expression of 53BP1 recurs in a variety of tumors and is associated with decreased survival without metastasis, these aspects may be clinically important. As has been shown for BRCA1- and BRCA2- mutated tumors, drugs targeting DDR may select for secondary mutations, which result in partial or complete restoration of protein function. These secondary mutations may lead to alternative splicing, losing the mutant exon, and causing a second frameshift mutation that restores the reading frame [45,46,47].

The newly discovered field of translesion synthesis (TLS) has suggested that mammalian cells need distinct polymerases to efficiently and accurately bypass DNA lesions. Translesion synthesis is the newest and less characterized pathway of DNA repair. It involves DNA polymerases that facilitate DNA replicationby efficiently bypassing various DNA lesions in a relatively error-free manner. TLS polymerase gene expression, mutation, and regulation are important for cancer etiology and treatment.These polymerases are enzymes with low fidelity when copying undamaged DNA; however, they copy damaged DNA with much higher efficiency compared to the replicative DNA enzymes [48]. These findings have led to a model proposing that, during DNA replication, the proofreading DNA polymerase gets stalled at sites of DNA damage, and it is replaced by a low-fidelity DNApolymerase. The perturbation due to not proofreading DNA polymerase by mutation or loss of expressionis expected to result in the accumulation of mutations in cells exposed to specific carcinogens; therefore, TLS genes and TLS gene variations represent attractive pharmacologic targets [48].

In the case of the DNA mismatch repair (MMR) system, other resistance mechanisms are involved. To maintain genome integrity, the MMR system is essential, and mutations in MMR genes (such as MLH1 and MSH2) can lead to a microsatellite instability (MSI) phenotype [49]. MMR deficiency is associated with tolerance to different cytotoxic chemotherapies (e.g., doxorubicin); furthermore, resistance to cisplatin and carboplatin has been reported to cause hypermethylation of MLH1. However, a subset of cancers characterized by high tumor DNA damage and high susceptibility to immune checkpoint inhibitors can be identified by the increased level of microsatellite instability. Interestingly, cancers associated with defective MMR do not display the mutational disparity between regions with high and low histone H3 lysine 9 methyl 3 (H3K9me3). This loss of mutational disparity in MMR-deficient cancers is due to increased mutation rates in regions lacking H3K9me3. Thus, the higher recovery of mutations in H3K9me3 domains observed in MMR-competent cancer cells is likely caused by reduced MMR repair rates in high H3K9me3 regions. A change in histone modifications can directly affect DNA damage repair efficiency since many histone modifications have been implicated in promoting or inhibiting the recruitment of specific repair proteins [50,51].

Maintenance of genomic stability in cells requires a tight regulation of histone modifications that accompany DDR. A long-standing question in the study of the cellular response to DNA damage is how the complex structural environment of chromatin is altered in the presence of DNA lesions [50]. Although DNA damage repair mechanisms have been extensively studied, the exact order of histone modifier and remodeler recruitment remains obscure. It seems more evident that a number of recruited factors regulate each other’s accumulation and activation, rendering the study of the specific function of each factor more difficult. The basic repeating unit of chromatin, the nucleosome, consists of DNA wrapped around an octamer of core histones, which is composed of two molecules each of the histones H2A, H2B, H3, and H4. Nucleosomal DNA is dynamically packaged to varying degrees, resulting in different levels of chromatin compaction ranging from the 10nm fiber to higher-order structures such as the condensed mitotic chromosomes. Heterochromatin plays a major role in the maintenance of the genome stability by repressing transposable elements including DNA transposons and retrotransposons (RTEs), further classified as LTR sequences that characterize endogenous retroviruses (ERVs) and non-LTR elements such as long or short interspersed elements. Propagation of RTEs in the genome has been recognized as a great source of genomic instability [51]. Heterochromatin is classically described as a condensed, densely stained region of DNA that contains few active genes but is enriched for repetitive sequences. Mammalian heterochromatin is characterized by high levels of the histone modifications H3K9me3 and H3K27me3 and low levels of histone acetylation. Heterochromatin is maintained by a dense array of specific chromatin-binding proteins, including members of the HP1 family (which bind to methylated H3K9), histone deacetylases (HDACs), and histone methyltransferases. The importance of chromatin organization in maintaining genomic stability is underscored by studies of sequencing of multiple cancer genomes; these studies have revealed that mutations accumulate at much higher levels in compact, H3K9me3-rich heterochromatin domains [52,53]. Furthermore, insertion and deletions are localized around nucleosomes, whereas mutations tend to cluster on the nucleosomal DNA [54], and both can be influenced by the presence of specific epigenetic modifications on the nucleosome [52]. Some of these differences in mutation rates may occur via negative selection or through protection of DNA from mutagens by association with nucleosomes. However, the elevated mutation rates in compact, transcriptionally silent heterochromatin domains imply that chromatin packing may impact the detection or repair of damage by the DDR machinery. That is, the ability of the DDR machinery to access the DNA can have a significant impact on genomic stability within specific regions. Indeed, heterochromatin dysfunction provokes genetic instability by inducing aberrant repeat repair, chromosome segregations errors, and replication stress. The therapeutic potential of combination therapy to cope resistance relies on the simultaneous targeting of several different hallmarks of cancer; chromatin remodeling and epigenetic factors may influence cancer response to treatment in ways that are complementary to the more common targets of apoptosis and cell proliferation.

## 5. DNA Damage and Immunotherapy

The human cell is subject to approximately 50,000 DNA lesions per day, the majority of these being single-stranded or DSBs [55,56,57]. Immediately, cells with DSB activate the DDR signal, and many proteins involved in DSB processing and repair accumulate in the damaged area. DSBs are revealed by the Mre11–Rad50–Nbs1 (MRN) and Ku70–Ku80 complexes, which in turn recruit ATM and DNA-dependent protein kinase catalytic subunit (DNA-PKcs), respectively [58,59]. Mostly, the target is the C terminus of the histone variant H2AX, whose derivative phosphorylated on serine 139 (S139) is referred to as γH2AX [55,56,57]. Then, γH2AX is boundby the tandem BRCA1 C-terminal domain (BRCT) domain of the DDR mediator protein MDC1 (mediator of DNA damage checkpoint 1). ATM-mediated phosphorylation near the DSB site is propagated through MDC1 via phosphate-dependent MRN-ATM recruitment, resulting in megabase-based γH2AX-MDC1 [60,61]. During mitosis, signals are generated by γH2AX-MDC1 to evaluate the health of cells through the immune system, thereby preventing the occurrence of malignant tumors. In response to DNA damage, natural killer cells and/or cytotoxic T lymphocytes have evolved to use their activated receptors to recognize upregulated ligands on the surface of precancerous/cancer cells. As previously mentioned, DNA is under the constant threat of damage from exogenous and endogenous sources [62,63,64]. Cellular metabolism products (e.g., reactive oxygen species) contribute to DSBs associated with stress replication or chromosome mis-segregation [65]. In this context, evidence has been accumulated that innate immunity can be activated by endogenous DSBs [66]. Indeed, chromosomal instability generates DNA fragments that are recognized by sensors such as cyclic GMP-AMP (cGAMP) synthase (cGAS) or IFI16 (gamma interferon-inducible protein 16), both able to bind cytosolic DNA and to induce interferons (IFNs) [67,68,69]. Specifically, cGAMP and IFI16 bind and activate the stimulator of gene interferon protein (STING), STING induces transcriptional activation of interferon regulatory factor 3 (IRF3) and nuclear factor-kappa activated B cell light chain enhancer (NF-kB) [70,71,72]. Secreted INFs bind to the heterodimeric IFN receptors IFNAR1/IFNAR2 or IFNGR1/IFNGR2 and activate Janus kinase 1 (JAK1) to phosphorylate the signal transducer and transcription activator (STAT) family members to induce IFN-stimulated gene expression (ISGs) (Figure 3). Numerous ISGs have been identified, but their roles are barely known because scarce data have been released on ISGs and DNA damage associated with chromosome mis-segregation [73]. There are other ways in which DNA damage can induce innate immune responses. For example, in response to etoposide-induced DNA damage, ATM may activate STING independently of cGAS, thus triggering NF-kB-dependent transcription [67]. OAS genes encode members of the oligoadenylate synthetase family. The encoded proteins are regulated by DDR signaling and synthesize 2′-5′ oligoadenylate. This latter activates latent RNase L which results in both viral and endogenous RNA degradation. RNase L is activated by dimerization occurring upon 2′-5′ oligoadenylate binding and results in cleavage of cellular RNA [66].

In liver cancer, the use of immune checkpoint inhibitors (ICIs) against PD-1 on T cells to improve the ability of the immune system to attack tumor cells has proven to be a promising target for cancer treatment [74,75,76]. However, a meta-analysis of three randomized phase III clinical trials that tested inhibitors of PD-L1 or PD1 in more than 1600 patients with advanced liver cancer revealed that immunotherapy did not improve survival in patients with non viral liver cancer [77]. In two additional cohorts, patients with nonalcoholic steatohepatitis (NASH)-driven liver cancer who received anti-PD-1 or anti-PD-L1 treatment showed reduced overall survival [77]. Response rates to ICI treatment vary widely among different cancers; many tumor indications fall within a response rate range of 20% to 40%. Thus, resistance or non response to ICIs remains a critical issue, and a greater understanding of the multiple factors that can contribute to resistance is necessary to increase the proportion of patients who stand to benefit from this form of cancer immunotherapy.

One of the most commonly used biomarkers for predicting response to ICIs is the intensity of PD-L1 expression on tumor cells, which is positively correlated with response to ICIs in several cancer settings [78]. Expression of PD-L1 by the tumor is important not only because it ensures, at the most basic level, that targetable PD-1 receptor–ligand interactions exist in the tumor microenvironment but also because PD-L1 expression correlates with parameters associated with immune activation in the tumor, including activated CD8^+^T-cell responses and antigen presentation [78]. Another well-recognized biomarker that predicts response or non response to ICIs is the neo-antigen burden of the tumor, associated with response to anti-PD-1 by several cancers [79,80]. Indeed, a high neo-antigen burden is acquired through DNA damage/genomic instability or deficient mismatch repair [81]. The common aspect that determines response to ICIs by different tumors including liver cancer is their high neo-antigen level, which leads to increased visibility of the tumor to the immune system and the development of a more potent antitumor T cell response following ICI treatment.

Given this contribution of neo-antigen burden to ICI efficacy, tumors with low mutational burdens, such as NASH-related liver and pancreatic cancers, might be expected to respond poorly to ICIs, and this is indeed the case. The importance of neo-antigen load is clear within subtypes of same cancer; in colorectal cancer, anti-PD-1 therapy was effective in patients with tumors exhibiting deficient mismatch repair, which demonstrated a 40% objective response rate, but not in patients with tumors with intact mismatch repair, which exhibited a non response [78]. Thus, as a general observation, tumor types exhibiting low neo-antigen levels are commonly associated with resistance or non responsiveness to ICIs. Therefore, low availability of neo-antigens is a major contributor to resistance to ICIs.A key feature of resistance is that the cell becomes accustomed to the presence of DNA damage; therefore, therapeutical strategies aimed at combining DNA-damaging drugs with ICIs need to be considered with caution.

To elude immunity, cancer cells can silence cGAS, IFI16,STING, HLA, and PD-L1 expression [70,82]. Given its key function in controlling the expression of PD-1 ligands and emerging evidence of its role in acquired resistance to immune checkpoint inhibitors, there is a renewed interest in IFN signaling. The main clinical finding is that antitumor immunity is also caused by abolishing the DDR protein BRCA1 or BRCA2 that is known to induce DSB accumulation [83]. In addition, overexpression of STING, TBK1, and IRF3-dependent chemokines CCL5 and CXCL10 has been observed in human cells lacking BRCA1 or BRCA2. Two subsequent studies [84,85] determined the extensive transcriptional characteristics of innate immune genes based on the cGAS–STING pathway in cancer cells lacking BRCA1 or BRCA2. These results show that the lack of BRCA1 or BRCA2 due to genome instability can lead to high levels of endogenous DNA damage, thereby promoting innate immune responses. However, these BRCA1/2-deficient cancer cells have been able to develop tumors. Therefore, it has been studied whether increasing DNA damage with drugs can improve innate immunity. Indeed, tumors lacking BRCA1/2 are hypersensitive to PARP inhibitors, which act by inhibiting the catalytic activity of PARP and relying on DNA repair mechanisms, resulting in the death of cells lacking BRCA1/2, but not that of normal cells [34,36]. Studies in BRCA1/2-deficient cells and tumors reported that treatment with PARP inhibitors (i.e., niraparib) increased endogenous DNA damage and genomic instability, inducing cGAS–STING activation and, hence, a reliable persistent immune response [86,87,88].

## 6. Perspectives

Despite the increase in resistance to cancer treatments, we should not lose sight of the fact that chemotherapy and molecularly targeted therapies have led to much improved survival for patients. Most importantly, we need to be able to stratify patients on the basis of whether they are likely to respond to specific drugs or drug combinations, especially after the introduction of immunotherapy. The use of powerful high-throughput technologies (such as next-generation sequencing) provides a large amount of data that can be used to identify potential predictive biomarkers for patient stratification. Cell lines are the best platform for identifying clinically relevant biomarkers and evaluating drug combinations. However, although cell lines are a good starting point, it is clear that improved in vitro and in vivo models are needed to more accurately assess drug resistance, evaluate potential drug combinations, and determine the therapeutic utility of predictive biomarkers. Next, such preclinical research needs to be tested in the clinic, which will require the design of trials that incorporate next-generation sequencing technology. Combined with interventions designed to increase CIN levels, targeting CIN resistance mechanisms may be an effective way to reduce the acquisition of resistance, and it may also enhance the response to other therapies that occasionally increase CIN levels.

## Figures and Tables

**Figure 1 ijms-24-00245-f001:**
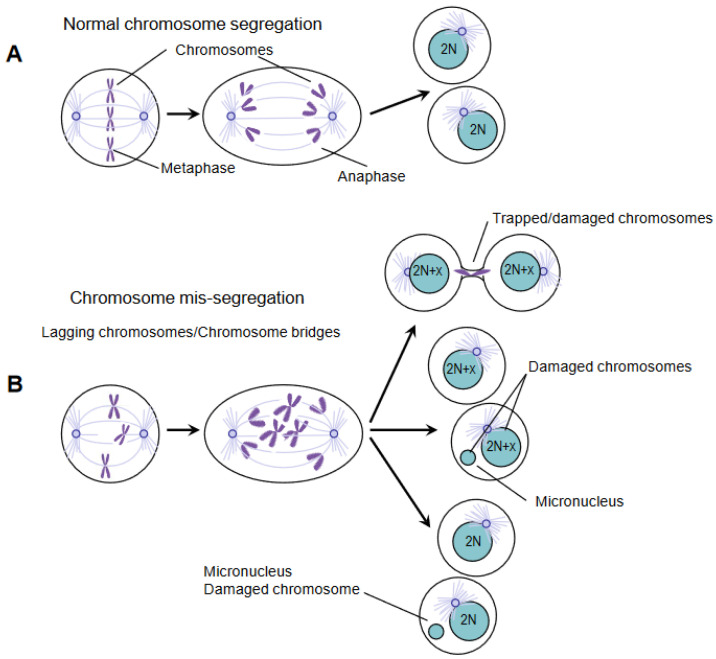
Chromosome mis-segregation causes DNA damage. (**A**) Precise chromosome segregation leads to the equal partitioning of the genome and the generation of two euploid daughter cells with diploid karyotype (illustrated as 2N). (**B**) Defective chromosome segregation can have multiple fates. Chromosomes can be trapped in the cytokinetic furrow and break during cytokinesis (**top**). Alternatively, chromosomes can mis-segregate and form micronuclei (**middle**), or they can be accurately segregated in nucleus/micronucleus in the daughter cells (**bottom**). Irrespective of how micronuclei are generated, their DNA is significantly damaged and elicits a DNA damage response.

**Figure 2 ijms-24-00245-f002:**
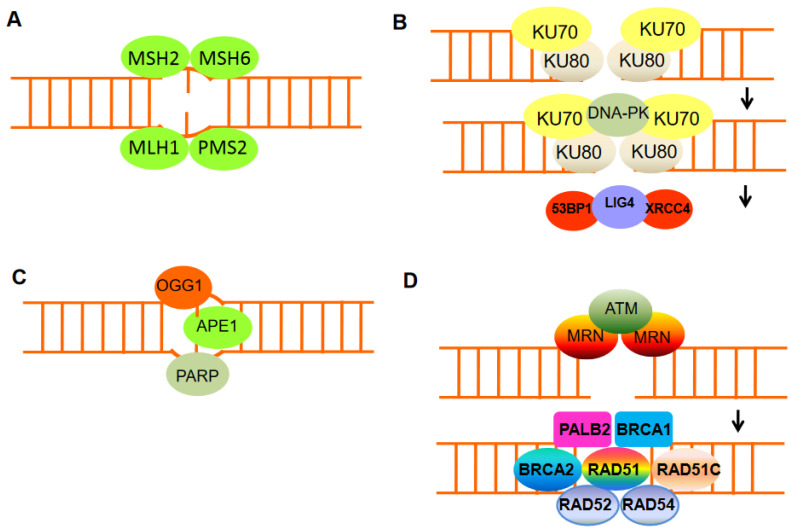
The mechanisms of DNA damage repair. (**A**) Mismatch repair proteins localize and repair mismatched bases. (**B**) Nonhomologous end-joining (NHEJ) requires a nuclease to resect damaged DNA, DNA polymerases to fill in new DNA, and a ligase to restore integrity to the DNA strands. The Ku bound to the end of the DNA can be considered as a Ku:DNA complex, which is a node where the nuclease, polymerase, and ligase of NHEJ can dock. DNA-PK has a molecular weight of 469 kDa and contains 4128 aa. It is the largest protein kinase in biology and is the only protein kinase that is specifically activated by binding to the ends of double-stranded DNA with multiple end configurations. Activated DNA-PK stimulates the ligase activity of XRCC4: DNA ligase 4. (**C**) Base excision repair can remove damaged DNA bases, e.g., by oxidation, using damage-specific DNA glycosylase, in the case of oxidizing guanine, 8-oxoguanine DNA glycosylase 1 (OGG1). (**D**) DNA double-strand break repair by homologous recombination (HR) can only be active in late S or G2 phases of the cell cycle when sister chromatid templates are available. HR includes DNA excision through MRN, homologous pairing and strand exchange through RAD51 and RAD52, Rad51c, Rad54, and BRCA1–PALB2, and DNA synthesis.

**Figure 3 ijms-24-00245-f003:**
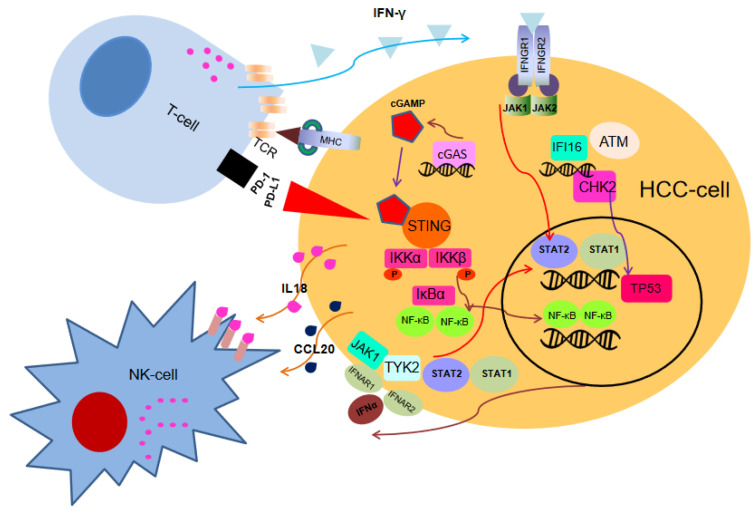
DNA damage promotes immune response. Endogenous or exogenous genotoxic stress triggers the accumulation of DNA damage, thereby forming DNA fragments. The DNA sensor cGAS binds to the DNA and uses GTP and ATP to catalyze the synthesis of cGAMP. cGAMP activates the endoplasmic reticulum (ER)-bound STING. NF-κB released from IkBa is activated and translocates into the nucleus, where it induces transcriptional targets, including cytokines. The secreted IFNs binds to their heterodimeric receptor IFNAR1/2 or IFNGR1/2 and activates JAK phosphorylation of STAT1 and STAT2. The STAT1/2 heterodimer acts as a transcription factor and triggers the expression of cytokines (such as IL18, IL15, CCL20). STAT1/2 heterodimer acts as a transcription factor and triggers the expression of cytokines such as IL18, IL15, and CCL20.Cytokines activate immune effector cells (i.e., natural killer cells and T cells) by promoting tumor antigen/neoantigen presentation on the major histocompatibility complex I (MHC-I) and PD-1/ PD-L1 expression.

## Data Availability

Not applicable.

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
