# Peer review of "The Adaptability of Chromosomal Instability in Cancer Therapy and Resistance"

_ijms, 2022, doi:10.3390/ijms24010245_

Round 1

Reviewer 1 Report

The authors nicely described very important topic on tumor therapy and DNA/chromosome damage. Review is clearly written with graphic suplement. I have no substantial remarks. Only suggestion - to check and unify  the references. 

Author Response

Reviewer No. 1

We thank this reviewer to consider our article interesting and informative. We have re-write the section where we quoted the mechanisms  of mis-segregation for a better comprehension

Author Response

Reviewer No. 2

 We also thank this reviewer for her/his time and effort in reviewing our manuscript

Reply to specific comments follows:

We have modified abstract adding types of vulnerabilities.

We have included information about Shieldin and TLS as suggested

We have amended all typos   errors introduced by  word software

Line 232 I suggest insertion and not inserts.

Amended

Line 276 reference for that statement of mitosis, there is a difference between sensing abnormal mitosis and micro-nuclei formation. Please explain.

Amended

Line 376 not clear please rephrase

Amended

Reviewer 3 Report

1. Correct the spacing and punctuation throughout the document. 

2. Please correct and rewrite the sentences in Lines 32 - 34, 86 - 88, and 94 - 96.

3. Please put down the reasons responsible for defective chromosomes' segregation in a numbering format for easy understanding.  

4. Please correct the conjoined words in Lines: 155, 165, 211, 223 - 226, 240, 316, and 334.

Author Response

Reviewer No. 3

We also thank this reviewer for her/his time and effort in reviewing our manuscript

We have amended all typos   errors introduced by  word software

Please put down the reasons responsible for defective chromosomes' segregation in a numbering format  for easy understanding

We have re-write the section where we mentioned the mechanisms  of mis-segregation for a better comprehension

Round 2

Reviewer 2 Report

The review is  significantly better. The authors addressed all the concerns.